# On the Use of Satellite Observations to Fill Gaps in the Halley Station Total Ozone Record

Lily N. Zhang[1], Susan Solomon[1], Kane A. Stone[1], Jonathan D. Shanklin[2], Joshua D. Eveson[2], Steve Colwell[2], John P. Burrows[3], Mark Weber[3], Pieternel F. Levelt[4,5], Natalya A. Kramarova[6], David P. Haffner[6,7]

[1]Earth, Atmospheric, and Planetary Sciences, Massachusetts Institute of Technology, Cambridge, 02139, USA
[2]British Antarctic Survey, Cambridge, CB3 0ET, UK
[3]Institute of Environmental Physics, University of Bremen, Bremen, 28334, Germany
[4]KNMI, De Bilt, 3731, The Netherlands
[5]University of Technology Delft, Delft, 2628, The Netherlands
[6]NASA Goddard Space Flight Center, Greenbelt, 20771, USA
[7]Science Systems and Applications, Inc., Lanham, 20706, USA

*Correspondence to*: Lily N. Zhang (lnz0018@gmail.com)

**Abstract.** Measurements by the Dobson ozone spectrophotometer at the British Antarctic Survey's (BAS) Halley research station form a record of Antarctic total column ozone that dates back to 1956. Due to its location, length, and completeness, the record has been, and continues to be, uniquely important for studies of long-term changes in Antarctic ozone. However, a crack in the ice shelf on which it resides forced the station to abruptly close in February of 2017, leading to a gap of two ozone hole seasons in its historic record. We develop and test a method for filling in the record of Halley total ozone by combining and adjusting overpass data from a range of different satellite instruments. Comparisons to the Dobson suggest that our method reproduces monthly ground-based total ozone values with an average difference of 1.1 ± 6.2 DU for the satellites used to fill in the 2017-2018 gap. We show that our approach more closely reproduces the Dobson measurements than simply using the raw satellite average or data from a single satellite instrument. The method also provides a check on the consistency of the provisional data from the automated Dobson used at Halley after 2018 with earlier manual Dobson data, and suggests that there were likely inconsistencies between the two. The filled Halley dataset provides further support that the Antarctic ozone hole is healing, not only during September, but also in January.

## 1 Introduction

Using the Halley Dobson record, Farman et. al. (1985) were the first to identify the austral springtime Antarctic ozone hole, a discovery that would change the fundamental scientific understanding of atmospheric ozone chemistry and contribute to environmental policy at the international level via the Montreal Protocol (Birmpili, 2018). The length of the Halley Dobson record as well as Halley station's particular location relative to the polar vortex and solar terminator have made it not only historically important but also uniquely valuable to modern studies of Antarctic total ozone.

In 2017, this remarkable record was interrupted. That February, Halley station was forced to cease operations due to risks associated with the structural stability of the Brunt ice shelf upon which it rests (https://www.bas.ac.uk/media-post/halley-

research-station-antarctica-to-close-for-winter/). No ozone data were taken during the austral springs of 2017 or 2018, breaking the continuity of this unique record of the springtime ozone hole. The measurement season at Halley typically spans August through April of each year (although there are a few missing months in years before the ice crack issue, discussed further below). No routine ozone data are available at Halley in the Antarctic winter months of May, June, and July, when the sun is below the horizon. Halley is now only staffed during the Antarctic summer season, with automated instrumentation operating

throughout the measurement season, including the automated Dobson instrument. The transition from manual year round operation to automated operation is reflected in the post-2017 change in seasonal coverage in the Halley ozone record shown in Fig. 1 (which also shows satellite data for comparison, discussed further below).

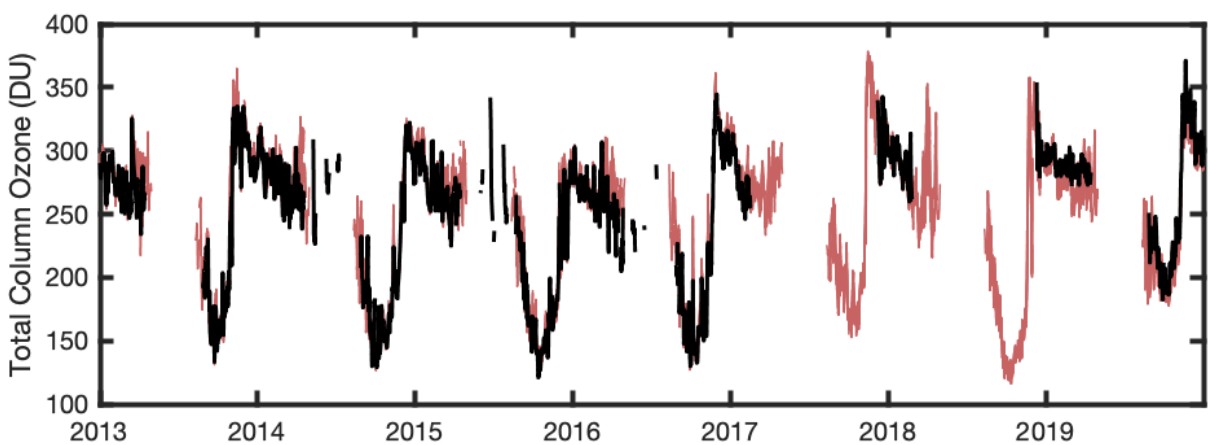

**Figure 1: Daily averages for total column ozone measurements by Dobson instruments at Halley station (in black) overlaid on top of all available (raw) satellite daily averages (in red) from 2014-2019.**

In the first decades of the satellite observing system, overpass comparisons with the ground-based Dobson network were used for validation: e.g., to identify problems with different satellite systems such as calibration drifts or performance under cloudy conditions (Bojkov et al., 1988; McPeters and Labow, 1996). As the satellite observing system matured, satellite/Dobson comparisons could be used in the opposite sense: for example, to find particular Dobson stations that were inconsistent with

the rest of the ozone observing system (e.g., Fioletov et al., 1998). Therefore, we undertook the development of an approach to fill in missing periods in a specific Dobson ozone dataset using satellite data.

The recent gap in the Halley record limits its use for studying the full record of Antarctic ozone, particularly the current era of ozone healing, as global chlorofluorocarbon concentrations slowly decline. Satellite records of total ozone began in the 1970s (Heath et al., 1973) and provide complementary information, with shorter data records than those of the historic ground-based

stations such as Halley, but complete global coverage and routine day-to-day observations. Here we examine a technique to combine satellite Halley overpass observations from a variety of different available satellite instruments to provide as complete a record of Halley total ozone as possible. Using satellite data, we develop and test a method to fill in the record of Halley total

ozone as would have been measured by the Dobson instrument. Our goal is not to obtain the "most accurate" value for total ozone over Halley, but rather to reproduce what the Dobson instrument would have observed, had it been in operation. We focus on the gaps from 2017 to 2018, but also apply the method where possible to fill in missing months in the earlier historical data.

## 2 Methods

### 2.1 Data

All Halley Dobson data were obtained directly from the British Antarctic Survey (https://legacy.bas.ac.uk/met/jds/ozone/index.html#data). Halley solar data typically end on April 16[th] as the sun retreats for polar night, and resume on August 27[th]. There are also some limited lunar measurements. For observations between 1956 and 1971, only daily averages are currently available. Provisional individual Dobson measurements of total column ozone at Halley are available from 1972 onwards and were used to compute daily averages. Data from the automated instrument for 2018 onwards are particularly likely to require revision as cross-calibration only takes place during the short summer season.

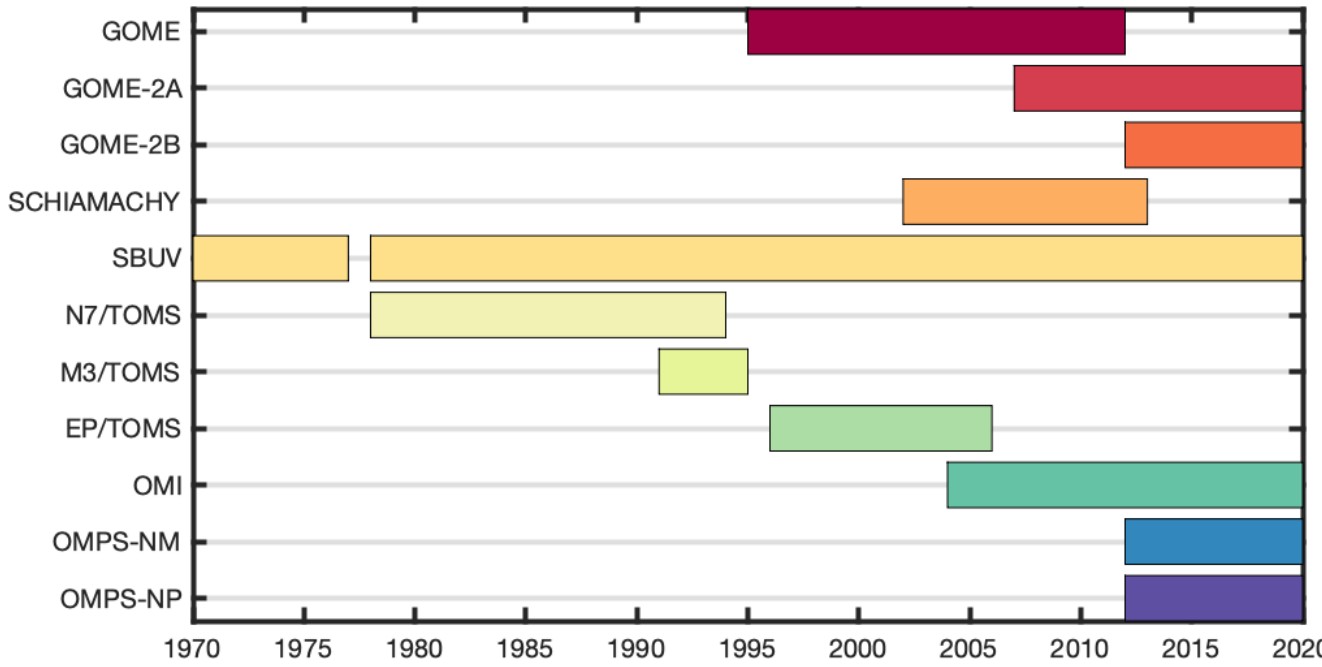

**Figure 2: Timeline showing years with available measurements from each satellite instrument considered for filling the gaps of the Halley Dobson total ozone record.**

The different satellite datasets use a variety of spectral ranges, scan widths, ozone absorption cross sections, and retrieval algorithms to determine total ozone. In this study, we analyze Halley overpass data from the following eleven instruments (Fig. 2): GOME (Global Ozone Monitoring Experiment), GOME-2A, GOME-2B, SCIAMACHY (SCanning Imaging Absorption spectroMeter for Atmospheric CartograpHY), SBUV (Solar Backscatter Ultraviolet), N7/TOMS (Total Ozone Mapping Spectrometer on Nimbus-7), M3/TOMS (Meteor-3), EP/TOMS (Earth Probe), OMI (Ozone Monitoring Instrument), OMPS-NM (Ozone Mapping and Profiler Suite, Nadir Mapper), and OMPS-NP (Nadir Profiler). Comparison of the satellite overpass data with Halley Dobson over 2013–2019 is shown in Figure 1, highlighting the missing Halley Dobson data during the 2017 and 2018 austral springs. All instruments use only UV wavelengths in their ozone retrieval. Version numbers and data availability for each satellite instrument are listed below in Table 1.

**Table 1: Version numbers, sources, and URLs for each of the eleven satellite instruments used in the study. For the NASA GSFC instruments, we provide 2 numbers. The first one represents the version number for the algorithm. The second represents the date version. In some cases, the algorithm and data version are the same.**

| Instrument | Version | Source | URL |
|---|---|---|---|
| GOME | WFDOAS V1 | University of Bremen | https://www.iup.uni-bremen.de/UVSAT_material/data/ satellite_overpass_HalleyBay_Syowa/ |
| GOME-2A | WFDOAS V4 | University of Bremen | https://www.iup.uni-bremen.de/UVSAT_material/data/ satellite_overpass_HalleyBay_Syowa/ |
| GOME-2B | WFDOAS V4 | University of Bremen | https://www.iup.uni-bremen.de/UVSAT_material/data/ satellite_overpass_HalleyBay_Syowa/ |
| SCIAMACHY | WFDOAS V1 | University of Bremen | https://www.iup.uni-bremen.de/UVSAT_material/data/ satellite_overpass_HalleyBay_Syowa/ |
| SBUV | V8.6/V8.6 | NASA GSFC | https://acd-ext.gsfc.nasa.gov/anonftp/toms/ |
| N7/TOMS | V8.0/V8.6 | NASA GSFC | https://acd-ext.gsfc.nasa.gov/anonftp/toms/ |
| M3/TOMS | V8.0/V8.0 | NASA GSFC | https://acd-ext.gsfc.nasa.gov/anonftp/toms/ |
| EP/TOMS | V8.0/V8.6 | NASA GSFC | https://acd-ext.gsfc.nasa.gov/anonftp/toms/ |
| OMI | V8.0/V8.5 | NASA GSFC | https://acd-ext.gsfc.nasa.gov/anonftp/toms/ |
| OMPS-NM | V8.0/V2.1 | NASA GSFC | https://acd-ext.gsfc.nasa.gov/anonftp/toms/ |
| OMPS-NP | V8.6/V2.6 | NASA GSFC | https://acd-ext.gsfc.nasa.gov/anonftp/toms/ |

The European GOME (Global Ozone Monitoring Experiment; see Burrows et al., 1999) and GOME-2 sensors (since 2007; Munro et al., 2015) are nadir sounding instruments while SCIAMACHY (SCanning Imaging Absorption spectroMeter for Atmospheric CartograpHY; 2002-2012) is a combined limb-, occultation-, and nadir-viewing spectrometer (Bovensmann et al., 1995), all with a common heritage (Burrows et al., 1995). The total ozone columns from GOME, SCIAMACHY nadir, GOME-

2A and GOME-2B are retrieved using the weighting function differential optical absorption spectroscopy (WFDOAS) technique in the spectral window 325-335 nm (Coldewey-Egbers et al., 2005; Orfanoz-Cheuquelaf et al. 2021). The WFDOAS approach was validated using Halley station data as reported in Weber et al. (2005) and Orfanoz-Cheuquelaf et al. (2021). The minimum footprints (ground pixel sizes) are 320 km by 40 km for GOME, 60 km by 30 km for SCIAMACHY nadir, and 80 km by 40 km for both GOME-2 sensors. Daily mean overpasses were calculated by averaging ozone columns from all ground

pixels within 100 km (GOME-2) and 300 km (SCIAMACHY, GOME) of the station.

The Solar Backscatter Ultraviolet (SBUV) record is the longest satellite record and includes measurements from 9 satellite instruments starting from the Backscatter Ultraviolet (BUV) on Nimbus-4 followed by the Solar Backscatter Ultraviolet (SBUV) instrument on Nimbus-7 and a series of SBUV/2 sensors on NOAA-9, 11, 14, 16, 17, 18, and 19. The SBUV instruments measure Earth's radiance at discrete wavelengths in the spectral range from 252 to 340 nm, with a spatial field of

view of about 170 km x 170 km at the surface. These measurements have been cross-calibrated (DeLand et al., 2012) and processed with the same retrieval algorithm (Bhartia et al., 2013) to produce a consistent, climate-quality record of ozone profiles and total columns (Frith et al., 2014). The method for creating overpasses for SBUV is described by Labow et al. (2013, see Sect. 5 there).

The Total Ozone Mapping Spectrometer (TOMS) on Nimbus-7 provided the first maps of total ozone over Antarctica from

space (Stolarski et al., 1986; Bhartia and McPeters, 2018). Two additional TOMS instruments were later launched on the Meteor-3 (M3) and Earth Probe (EP) satellites. The TOMS instruments made measurements at discrete wavelengths in the spectral range from ~309 to 380 nm with a spatial resolution of about 50 by 50 km at nadir and increase to 150 by 200 km at the extreme cross-track positions.

The Dutch-Finnish Ozone Monitoring Instrument (OMI) is a nadir-looking, push broom UV/Visible solar backscatter

spectrometer on NASA's Aura satellite that measures the Earth's radiance spectrum from 270 to 500 nm with a spatial resolution of 13 km x 24 km at nadir and approximately 125 x 125 km at the outermost scan positions (Levelt et al., 2006). The OMI total ozone dataset used here is produced with a variation of the same algorithm used for the TOMS instruments and validation of the record has shown OMI to be stable for studies of ozone trends (McPeters et al., 2008, 2015).

OMPS-NM and OMPS-NP are both from the Ozone Mapping and Profiler Suite on board of Suomi National Polar Partnership

(NPP) satellite. The OMPS Nadir Mapper (NM) has a wide swath to provide global daily maps of total ozone columns with a spatial resolution at nadir of 50 x 50 km. The OMPS Nadir Profiler (NP) sensor measures the complete spectrum from 260 nm to 310 nm and in combination with OMPS Nadir Mapper enables profile and total ozone retrievals for nadir direction only with a spatial resolution of 250 x 250 km at the ground (McPeters et al., 2019, Kramarova et al., 2014).

Overpasses for the TOMS, OMI, and OMPS-NP instruments are defined by selecting the single pixel most nearly co-located

with Halley Station. In the case of there being multiple pixels available, a pixel with high optical path will be rejected in favor of one with slightly poorer spatial coincidence but lower optical path. For the OMPS-NP instrument, the pixel closest to the station is chosen. None of these instruments, as well as SBUV, were validated with Halley station data.

Below, we first focus on the following six instruments: GOME-2A, GOME-2B, SBUV, OMI, OMPS-NP, and OMPS-NM. all of which were in operation during the period from 2013 to 2020 (spanning the period of missing Halley data from 2017 to 2018). We then include other instruments as appropriate for other periods. As with the Dobson data, individual overpass data of total column ozone were used to compute daily averages.

## 2.2 Data Analysis

From the individual satellite instruments, a "satellite average" daily total column ozone dataset was constructed, which represents the mean of all available satellite daily averages for each day.

Absolute and relative differences between satellite data with respect to the Halley Dobson were computed using daily values for each satellite individually, from which the satellite average was obtained. All comparisons and difference calculations were only considered on coincident days of satellite and Dobson measurements.

With all measurements and differences in the form of averaged daily values, data were categorized and then averaged according to their corresponding month and day of year (DOY). Months directly bordering the polar night (April and August) contained fewer data points when computing monthly averages.

Initial comparisons revealed the value of our method for identifying outliers in the Dobson data. In particular, lunar Dobson measurements from August 24th, 2015 were excluded due to obviously anomalous differences compared to satellite values observed on that day.

## 2.3 Delta Characterization and Adjustment

Biases between Halley and satellite data were characterized individually for each instrument by day of year, over the entire period of available observations. Note that the use of the word "bias" is not meant to imply an error, but rather a difference relative to the Halley Dobson. To avoid confusion, we will henceforth use the Greek letter Δ to denote this difference. Using only coincident days, the Δ value for each day of year is the average of the absolute differences between each satellite and Dobson for that day of year, across all years in each satellite series. Relative differences were also computed but displayed the same seasonality as absolute differences. To provide the value that would be seen by the Dobson, the corresponding Δ was then subtracted from each satellite's daily average. The delta-adjusted satellite average is the mean after each instrument's dataset has been individually delta-adjusted. Uncertainty for the delta-adjustment of the satellite average was calculated by combining, in quadrature, the standard error of the mean for each satellite and accounts for interannual variability.

## 2.4 Filling in Missing Halley Data

Daily Dobson measurements at Halley typically begin in the last week of August and end in the third week of April (August 27th to April 16th). For months when Dobson observations are not available, the delta-adjusted satellite average was used to fill

in daily averages for the days that Halley would typically be in operation. No attempt was made to fill in individual missing days within months for which Dobson data do exist, but rather only those months when Halley measurements are lacking.

## 3 Results and Discussion

**Table 2: Average absolute differences in DU between the total column of O3 retrieved from the Halley Dobson instrument and those retrieved from the (raw) daily measurements by GOME-2A, GOME-2B, OMI, OMPS-NM, OMPS-NP, SBUV averaged by month and in total for the period from 2013-2018.**

| Month | GOME-2A | GOME-2B | SBUV | OMI | OMPS-NM | OMPS-NP | Satellite Average |
|---|---|---|---|---|---|---|---|
| January | 0.3 | 4.1 | 6.5 | 4.8 | 4.6 | 7.3 | 4.5 |
| February | -1.5 | -0.1 | 3.3 | 2.8 | 1.6 | 5.4 | 1.8 |
| March | 7.9 | 11.4 | 6.0 | 6.6 | 2.3 | 6.0 | 6.7 |
| April | 9.2 | 17.7 | 24.3 | 8.6 | 7.8 | 24.3 | 17.7 |
| August | N/A | 10.1 | 12.3 | 7.6 | -4.7 | 12.3 | 6.2 |
| September | 2.8 | 6.3 | -0.5 | -0.6 | -2.5 | -0.7 | 1.0 |
| October | 2.1 | 4.2 | 2.4 | 4.9 | 4.4 | 3.5 | 3.5 |
| November | 0.7 | 4.8 | 5.6 | 6.1 | 6.4 | 6.2 | 4.9 |
| December | 2.5 | 6.4 | 6.8 | 3.8 | 4.4 | 7.1 | 5.1 |
| Total | 2.2 | 5.7 | 5.8 | 4.4 | 3.3 | 6.5 | 4.9 |

Average absolute difference values provide a measure of how the satellite data compare to the Dobson instrument (Table 1).

On average, GOME2A-, OMPS-NM, and OMI exhibit the lowest average difference with the Dobson of the individual instruments while the OMPS-NP instrument has the highest. Initial comparisons revealed that the use of the Serdyuchenko ozone absorption cross sections (Serdyuchenko et al., 2014) in the current GOME-2 data analysis method resulted in a 2-3% positive bias in total ozone when compared to the Bass and Paur cross sections (Paur and Bass, 1985) employed at Halley. For comparability with the other values, we adjusted GOME-2 data by a first order factor of 1.025 to account for the differences

in absorption cross sections before performing the above analysis. OMI is the only one out of the six displayed to use the Bass-Paur ozone absorption cross sections in its retrieval algorithm. The other NASA instruments—OMPS-NP, OMPS-NM, and SBUV—all use the Brion-Daumont-Malicet (BDM) cross sections (Malicet et al., 1995). While a scaling factor could be applied to adjust for the different cross sections used as was done for GOME-2, differences between OMPS-NM and OMPS-NP datasets would remain. The average of all satellite instruments consistently performs well relative to the individual

instruments in all months except April (see below), and in particular during the austral spring months of August, September and October. This supports the use of the satellite average for this study and application.

All Δ values were then applied by day of year in each individual satellite dataset for all periods of observations. Multiple instruments were averaged for each period whenever available, in the manner discussed above, and used to form the best available delta-adjusted satellite averages over time throughout the record.

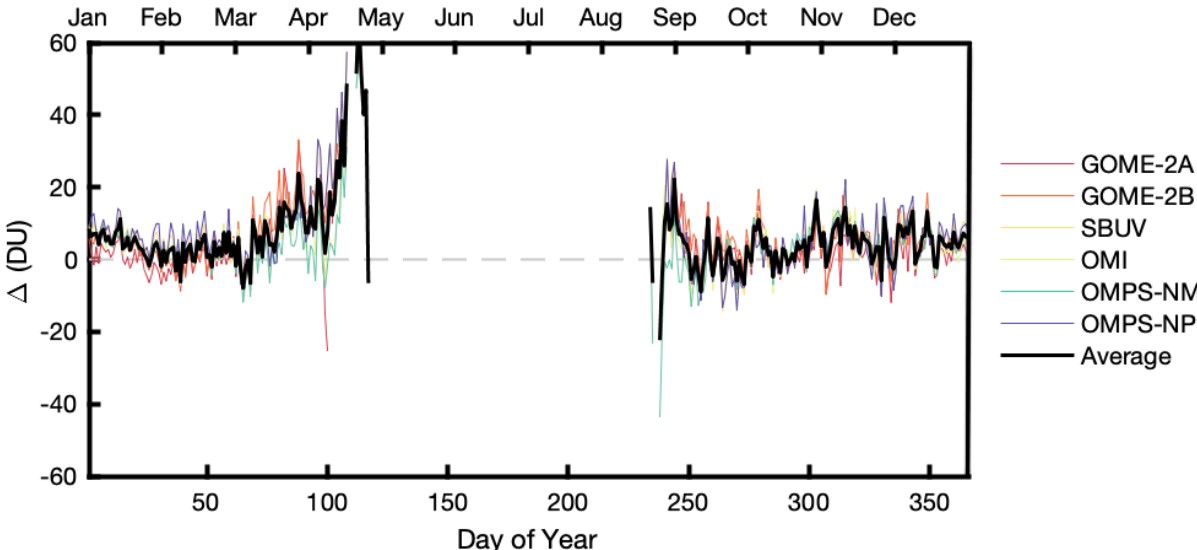

**Figure 3: Average Δ (over 2013-2018) between total O3 column retrieved from the measurements of the Halley Dobson and each satellite instrument by day of year, as well as the Δ averaged across all instruments.**

Characterizing Δ by day of year reveals trends across all instruments. Figure 3 shows that Δ is largest in the months of April and August, when solar zenith angles are large, as the station approaches and exits the polar night. The rapid and non-linear increase in Δ during spring and fall demonstrates the importance of defining the Δ in these seasons by average daily, rather than monthly differences. Additionally, Δ does not follow a simple solar zenith angle dependence. Values differ between the onset and end of the polar night for days with the same solar zenith angle, as evidenced by the larger Δs in April versus August. Therefore, we chose to characterize Δ by day of year rather than zenith angle.

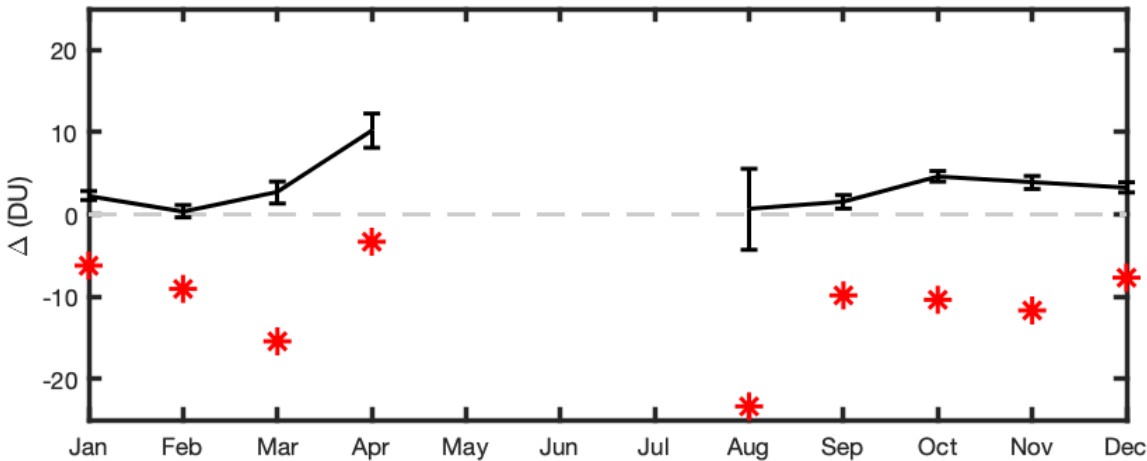

**Figure 4: Average Δ over all years (Fig. 2) excluding 2019 for each month with error bars (black). The monthly Δ values with the automated Dobson in 2019 (red) have larger magnitudes than Δs in other years. The error bars represent the standard error of each satellite mean, combined in quadrature for each monthly bin.**

Figure 4 reveals that the provisional 2019 automated Dobson displayed substantially larger negative Δ values compared to the rest of the dataset (Fig. 4). This indicates likely inconsistencies between the automated instrument and earlier data. Every Dobson instrument must be carefully calibrated to ensure accurate data; the calibration process for the automated instrument has not yet been completed. Therefore, we chose to exclude 2019 from our delta adjustment. Because the station continued to use the automated instrument in 2020, we treated the 2020 data as likely inconsistent as well and excluded it from our Δ adjustment. Figure 4 illustrates the value of our method for testing Dobson measurements for potential inconsistencies, particularly following instrument changes when calibration procedures may still be underway.

To test the fidelity of our method, we then omitted Halley Dobson measurements for selected time frames during which data were available and evaluated how well our method could reproduce those values. In short, after excluding the selected years, instruments were "trained" over the rest of the available range for the satellite (see Fig. 2) by determining the average Δ for each day of year between each of the satellites and Halley. We then applied that Δ to the satellite data for the omitted period to define what the delta-adjusted satellite average suggests that Halley should have observed. These values were then compared to what the Halley Dobson actually observed. We were particularly interested in evaluating our method for a time frame when the same satellite instruments as the ones in operation from 2017 and 2018 were available. Consequently, we chose to test the method for the years 2013 to 2015 by pretending data for those years did not exist and characterizing the monthly Δ values averaged over those years using the rest of the available data for the GOME-2A, GOME-2B, OMI, OMPS-NP, OMPS-NM, and SBUV instruments. To examine the performance of our method during periods when there were fewer available instruments, we also tested on 1998-2002 using data from GOME, SCIAMACHY, SBUV, and EP/TOMS instruments. The range of available data for each instrument can be found in Figure 2. The training period for each instrument is the available range after excluding the years being tested (and 2019-2020).

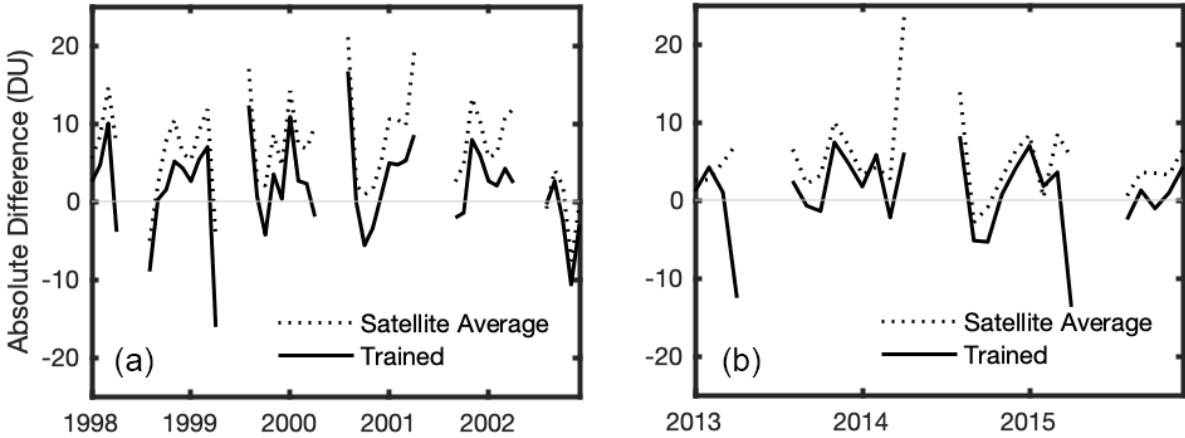

**Figure 5: The monthly mean of the absolute difference between the ozone columns, retrieved from Halley Dobson daily ozone averages and the satellite average (dotted) as well as the difference between the trained satellite average (solid) and the Dobson observations for the periods (a) 1998-2002 and (b) 2013-2015.**

Figure 5 shows that, after excluding 2019 and 2020 data, applying the results of training to the satellite average reproduced Halley Dobson monthly total ozone values with an average and an estimated training error of $1.8 \pm 6.7$ Dobson units (DU) for the period from 1998-2002 and $1.1 \pm 6.2$ DU for the period from 2013-2015. The raw satellite average only reproduced Halley Dobson monthly total ozone values to within an average of 6.5 DU for 1998-2002 and 4.6 DU for 2013-2015. On average, the $\Delta$-adjusted satellite average displayed significantly smaller differences than the raw average without including $\Delta$ adjustment, showing that our method reproduced well what the Dobson would have observed compared to the performance of the satellite average.

Characterizing $\Delta$ values by month, rather than day of year, results in comparable accuracy (within 0.79 DU for 2013-2015) but decreased uncertainty ($\pm 2.2$ DU for 2013-2015) in reproducing Halley Dobson monthly total ozone values. This result is expected, given that the day-of-year-characterized $\Delta$ values, when averaged over a month, should resemble the monthly-characterized $\Delta$. The decreased uncertainty in the monthly-characterized $\Delta$ is due to the greater number of data points averaged in the $\Delta$ adjustment. The use of one characterization over the other should depend on the goal of a given study. When reproducing daily total ozone values, as we do in this paper, $\Delta$ values need to be characterized by day of year in order to capture rapid changes in SZA and, subsequently, total ozone in the early spring and late fall (Fig. 3).

The $\Delta$-adjusted satellite data were then used to complete the Halley Dobson record (Table 2), including not only the period of the ice crack but other months when Dobson data are occasionally missing. No satellite data exist prior to 1970, and in the early 1970s, only one instrument (Nimbus-4 BUV) is available to fill in certain months. Comparison between Table 2 and Fig. 1 shows which satellite instruments are available to fill in various periods.

**Table 3: Monthly total ozone averages at Halley. Bold italic indicates months with no available Halley Dobson observations or only provisional automated Dobson data, for which the delta-adjusted satellite average was used.**

| Year | Jan | Feb | Mar | Apr | Aug | Sep | Oct | Nov | Dec |
|------|-----|-----|-----|-----|-----|-----|-----|-----|-----|
| 1956 | NA | NA | NA | NA | NA | 315 | 313 | 371 | 360 |
| 1957 | 335 | 297 | 289 | 275 | 302 | 285 | 322 | 396 | 349 |
| 1958 | 333 | 302 | 282 | 257 | NA | NA | 306 | 351 | 380 |
| 1959 | 343 | 329 | 298 | NA | NA | NA | 303 | 304 | 341 |
| 1960 | 323 | 299 | 296 | NA | NA | 288 | 293 | 347 | 377 |
| 1961 | 320 | 304 | 305 | NA | NA | 268 | 309 | 333 | 345 |
| 1962 | 312 | 298 | 330 | NA | NA | NA | 323 | 382 | 378 |
| 1963 | 321 | 303 | 306 | 288 | 315 | NA | 301 | 349 | 352 |
| 1964 | 318 | 301 | 326 | 304 | 272 | NA | 310 | 402 | 358 |
| 1965 | 316 | 295 | 297 | NA | NA | NA | 274 | 299 | 336 |
| 1966 | 300 | 290 | 284 | 287 | NA | 289 | 308 | 339 | 346 |
| 1967 | 300 | 285 | 269 | NA | NA | NA | 315 | 359 | 334 |
| 1968 | 320 | 286 | 290 | 281 | 285 | 281 | 293 | 387 | 350 |
| 1969 | 313 | 291 | 282 | 246 | NA | 286 | 275 | 298 | 316 |
| 1970 | 306 | 286 | 269 | 259 | 309 | *274* | 275 | 357 | 346 |
| 1971 | 319 | 314 | 275 | 279 | *303* | 280 | 291 | 375 | 346 |
| 1972 | 317 | 301 | 301 | 314 | 305 | 266 | 296 | 377 | 351 |
| 1973 | 306 | 293 | 286 | 277 | 272 | 263 | 271 | 326 | 334 |
| 1974 | 307 | 275 | 262 | 242 | NA | 244 | 272 | 337 | 351 |
| 1975 | 320 | 275 | 279 | NA | NA | 267 | 303 | 309 | 338 |
| 1976 | 314 | 272 | 257 | *251* | NA | 265 | 283 | 326 | 335 |
| 1977 | 318 | 280 | 275 | 253 | 290 | 239 | 251 | 332 | 360 |
| 1978 | 310 | 305 | 282 | 253 | NA | 264 | 284 | 345 | 337 |
| 1979 | 295 | 283 | 278 | 283 | *265* | 232 | 263 | 323 | 352 |
| 1980 | 324 | 292 | 290 | *278* | 328 | 236 | 226 | 293 | 340 |
| 1981 | 299 | 280 | 253 | *268* | *278* | 241 | 237 | 285 | 326 |
| 1982 | 290 | 278 | 260 | *285* | *267* | 210 | 218 | 268 | 322 |
| 1983 | 308 | 292 | 278 | 266 | *253* | 228 | 195 | 289 | 325 |
| 1984 | 301 | 272 | 273 | 267 | *242* | 215 | 194 | 248 | 322 |
| 1985 | 301 | 269 | 263 | 245 | *247* | 217 | 185 | 215 | 304 |
| 1986 | 286 | 273 | 247 | 227 | 253 | 212 | 233 | 282 | 309 |

| | | | | | | | | |
|---|---|---|---|---|---|---|---|---|
| **1987** | 301 | 278 | 274 | 274 | 254 | 182 | 150 | 188 | 287 |
| **1988** | 286 | 264 | 271 | 265 | 242 | 207 | 216 | 312 | 323 |
| **1989** | 284 | 281 | 260 | 274 | 270 | 186 | 150 | 255 | 295 |
| **1990** | 290 | 266 | 254 | 254 | 259 | 173 | 173 | 207 | 246 |
| **1991** | 281 | 257 | 263 | 233 | 204 | 163 | 137 | 232 | 296 |
| **1992** | 271 | 283 | 281 | 257 | 185 | 152 | 147 | 206 | 270 |
| **1993** | 284 | 275 | 277 | 256 | 209 | 167 | 122 | 179 | 285 |
| **1994** | 278 | 264 | 255 | 284 | 197 | 152 | 126 | 217 | 316 |
| **1995** | 278 | 269 | 256 | 254 | 218 | 160 | 130 | 164 | 252 |
| **1996** | 261 | 249 | 246 | 226 | 173 | 155 | 148 | 181 | 260 |
| **1997** | 278 | 265 | 247 | 243 | 218 | 171 | 141 | 210 | 286 |
| **1998** | 267 | 262 | 264 | 255 | 221 | 162 | 140 | 183 | 255 |
| **1999** | 272 | 259 | 254 | 267 | 205 | 172 | 143 | 172 | 254 |
| **2000** | 281 | 258 | 250 | 256 | 179 | 151 | 137 | 267 | 299 |
| **2001** | 286 | 261 | 251 | 245 | 224 | 148 | 138 | 209 | 265 |
| **2002** | 283 | 263 | 246 | 250 | 228 | 213 | 224 | 329 | 306 |
| **2003** | 282 | 280 | 268 | 246 | 205 | 155 | 158 | 229 | 292 |
| **2004** | 277 | 271 | 262 | 242 | 242 | 173 | 191 | 222 | 282 |
| **2005** | 275 | 262 | 253 | 242 | 207 | 158 | 155 | 253 | 290 |
| **2006** | 281 | 269 | 272 | 255 | 221 | 147 | 137 | 181 | 275 |
| **2007** | 286 | 281 | 270 | 255 | 186 | 150 | 159 | 214 | 290 |
| **2008** | 291 | 274 | 282 | 263 | 203 | 151 | 145 | 180 | 244 |
| **2009** | 286 | 264 | 249 | 234 | 200 | 153 | 165 | 216 | 293 |
| **2010** | 293 | 275 | 254 | 267 | 222 | 188 | 184 | 222 | 271 |
| **2011** | 290 | 278 | 275 | 245 | *197* | 160 | 140 | 186 | 267 |
| **2012** | 284 | 262 | 252 | 243 | 209 | 175 | 179 | 302 | 310 |
| **2013** | 285 | 270 | 270 | 251 | 186 | 170 | 177 | 306 | 296 |
| **2014** | 292 | 279 | 265 | 255 | 205 | 173 | 148 | 195 | 294 |
| **2015** | 289 | 267 | 255 | 256 | 241 | 179 | 139 | 171 | 253 |
| **2016** | 274 | 261 | 258 | 234 | 213 | 175 | 155 | 245 | 307 |
| **2017** | 285 | 265 | *263* | *263* | *240* | *196* | *175* | *309* | 307 |
| **2018** | 297 | 280 | *263* | *255* | *208* | *165* | *132* | *214* | 300* |
| **2019** | *286* | *280* | *268* | *261* | *204* | *208* | *197* | *293* | *300* |

| 2020 | 293 | 281 | 275 | 265 | 235 | 176 | 138 | 182 | 226 |

* Manual observations with Dobson 31 from December 10-31. May not be representative of the full month.

Figure 6 presents plots of September and January monthly mean total ozone at Halley, now with missing months filled in, illustrating the value of our method. For September, the now-complete long record from Halley is suggestive of ozone recovery at a rate of $1.34 \pm 0.64$ DU yr$^{-1}$ ($p = 0.05$) post-2000, although caution must be taken before drawing conclusions using single station data, due to potential systematic shifts of the location of the springtime polar vortex over time that has been noted in previous work (Hassler et al., 2011; Lin et al., 2009; Grytsai et al., 2017) and possibly other factors. A low p-value ($p \leq 0.05$) for the regression indicates that the trend is unlikely to have occurred by chance. This figure also shows that post-2000 January data also displays a positive trend of $0.44 \pm 0.20$ DU yr$^{-1}$ ($p = 0.04$). January does not display such shifts in the vortex; indeed, the vortex is essentially dissipated in this summer month. Fioletov and Shepherd (2005) showed that summer season total ozone is correlated with that in spring. The long records in September and January taken together hence support the view that ozone recovery is occurring, and the figure demonstrates the application of our method towards future studies of long-term trends in Antarctic ozone.

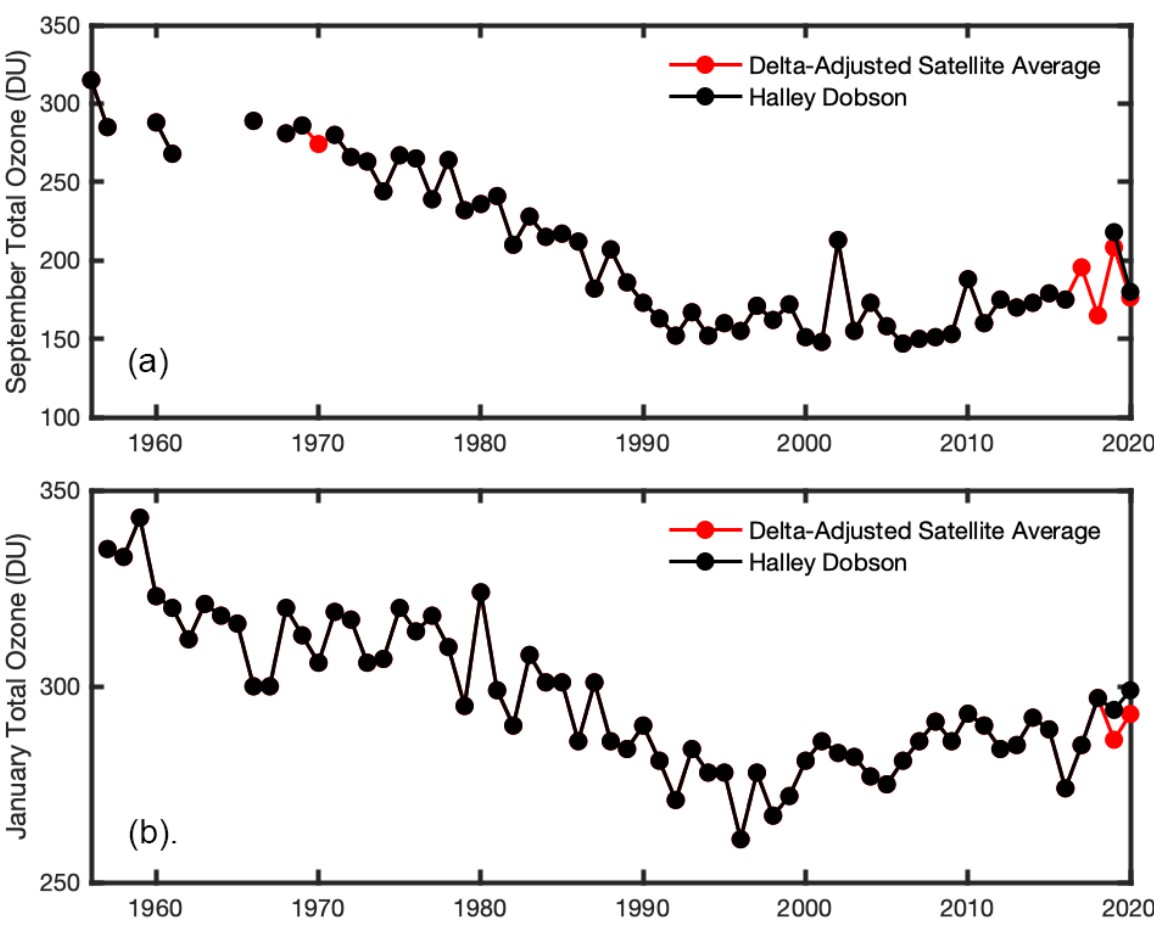

Figure 6: Monthly Halley ozone averages over time (black) for (a) September and (b) January, with the delta-adjusted satellite average (red) filled in for years with no or provisional Halley Dobson observations. Note that GOME and SBUV data are not yet available. Dobson data from 2019 and 2020 were replaced due to apparent inconsistencies between the automated instrument and earlier data.

## 4 Conclusions

We developed a method to fill in missing data in the historic Halley record of total ozone (Farman et al., 1985; Jones et al., 1995) using satellite overpass data, with a particular focus on the period of 2017-2018 when Halley station was abruptly closed for safety reasons associated with a crack in the ice shelf. We analyzed the suite of total ozone data from a range of available satellite total ozone instruments. Using the differences between daily Halley and satellite overpass data, we derived the differences ($\Delta$) between the Dobson and each satellite for each day of the observing season (August to April) as well as the satellite average. Through this process, we found that the preliminary computed data from the automated instrument in 2019 had apparent inconsistencies with the earlier data taken with the manual Dobson when compared to the satellite (see Fig. 4).

This comparison illustrates that our method can be valuable in identifying potential calibration issues, particularly after instrument changes.

We found that the average of the available satellites over 2013-2018 displayed a smaller Δ relative to the Halley total ozone data than most of the individual satellites and performed especially well during months in the austral spring. We then tested our method using time periods when Halley data were actually available to see how well the technique would have worked if data were missing at those times. Our tests indicate that by accounting for Δs between the daily satellite averages and Dobson data, we could fill in missing months with a high degree of fidelity (average difference of $1.1 \pm 6.2$ DU for monthly averages). We applied the method to all possible missing months of data in the Halley record, and the filled dataset will be available for use by other researchers.

The filled dataset allows study of the important question of the healing of the ozone hole due to the phaseout of new production of ozone depleting substances under the Montreal Protocol, which would otherwise be impeded by the years of the ice crack interruption. The results better support the conclusion that healing of the ozone hole is beginning in the key month of September than would be possible without the data filling, although we note that data for a single station in September can be influenced by changes in the position and conditions of the polar vortex, as documented in other studies. However, we also show that the Halley data indicate ozone healing for January as well, a month when the vortex is very weak and essentially circumpolar. Because of COVID-19, several Antarctic stations are currently subject to reduced operations and staffing (Hughes and Convey, 2020). The COVID-19 pandemic underscores that long-term observations may be unexpectedly interrupted at any time, due not only to geophysical change such as the ice crack but also societal change. The method developed here could be applied to bridge missing data in other station records.

## Code availability

MATLAB was used for data analysis and visualization. Scripts can be accessed at https://www.ssolomongroup.mit.edu/toolsandproducts

## Data availability

Sources for all data used in this manuscript can be found in Table 1. The filled Halley record shown in Table 3 is available for download at: https://www.ssolomongroup.mit.edu/toolsandproducts

## Author contribution

SS and KS conceptualized the project. The methodology was developed by SS, KS, and LZ and implemented by LZ with satellite and Halley Dobson data provided by the other co-authors. LZ prepared the manuscript with contributions from all co-authors.

## Competing interests

The authors declare that they have no conflict of interest.

## Acknowledgements

LZ acknowledges support by the Bacon Fund for undergraduate research. SS acknowledges support by the Lee and Geraldine Martin Chair in Environmental Studies at MIT. The research of JPB and MW is in part supported by the University and the State of Bremen, Germany, DFG (German Research foundation), DLR (German Aerospace) and BMBF (SynopSys). We thank EUMETSAT for providing level 1 data from GOME-2A and GOME-2B. Helpful discussions with Paul Newman are gratefully acknowledged.

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
