# Peer review of "On the Use of Satellite Observations to Fill Gaps in the Halley Station Total Ozone Record"

_Atmospheric Chemistry and Physics, 2021_

## Author Comment (AC1)

The authors would like to thank the reviewer for taking the time to provide feedback on "On the Use of Satellite Observations to Fill Gaps in the Halley Station Total Ozone Record." The insightful comments we received helped us understand how to improve on and better communicate the ideas presented in the paper. Nearly all suggestions were incorporated, and a line-by-line response can be found below (with author comments in blue):

**RC1: 'Interesting, but needs clarification of uncertainty measures', Anonymous Referee #1, 26 Mar 2021**

**Overall Comments**

The manuscript describes the use of satellite data of total ozone to fill gaps in the ground-based Dobson total ozone record at Halley Bay, Antarctica.

As mentioned in the text, Halley Bay has one of the longest and most important total ozone records. This record was the key for the dectction of the Antarctic ozone hole.

It is a good idea, and scientifically sound, to fill gaps in this important record with satellite data, as well as check for consistency. Overall, the paper is well written and merits publication in ACP.

Before publication, however, I suggest a few important clarifications. Thoroughout the manuscript, I get confused about the use of individual total ozone measurements, daily averages and difference, monthly averages and differences, and the corresponding standard deviations. Sometimes standard deaviations appear to be mis-named "averages" as well.  The description of the applied method to shift satellite data towards the Dobson data is also quite long-winded. It would benefit from shortening and clarification. There is no need to make a simple average bias correction appear much more complicated than it is.

**Specific Comments**

line 21, average of 2 Dobson Units: I don't think this is what is meant. My understanding is the each satellite record is shifted by the $\Delta$ from Fig. 3, so that it matches the Dobson data on average. Therefore the satellite average should reproduce the Dobson average exactly, by construction. What is probably meant here is "within a standard deviation of 2 Dobson units". Even more information is required here: are the (presumably) 2 DU standard deviation for monthly means or for daily means? Is the given value 1 or 2 standard deviations? Is it +-1 DU or +-2 DU? Is it even correct? In lines 184 and 185 the stated standard deviation of the differences is 6 to 7 DU. This is much larger than 2 and neesd to be checked.

Thank you for highlighting this issue, and we apologize for the lack of clarity. The original phrase, *"Tests suggest that our method reproduces the monthly ground- based Dobson total ozone values to within an average of 2 Dobson,"* was based on the results of testing how well the satellites used to fill in the 2017-2018 gap could reproduce Dobson monthly means using our $\Delta$-adjustment method (tested on 2013-2015). The phrase is indeed misleading. A more accurate description of our results would be that *"Tests suggest that our method reproduces monthly*

*ground-based Dobson total ozone values* **with an average difference of 1.1 ± 6.2 DU for the satellites used to fill in the 2017-2018 gap**," and we have made that change in the text.

The reason the Δ-adjusted satellite average does not reproduce the Dobson averages exactly in our tests is because the years being tested were not included in the calculations of the Δs. Rather than calculate Δs for 2013-2015 and subtract them from the 2013-2015 satellite data (which would lead to an exact reproduction), we pretended the 2013-2015 Dobson data was "missing" to see how well our method could fill in Dobson data by using Δs characterized from the rest of the available data—as would be the case for filling in genuine data gaps, as in 2017-2018. Although the Δs would most likely not be exactly the same as those of the years being tested, our results showed that our Δ-adjustment method was still able to improve on the raw satellite average. This allowed us to understand how well we could fill in the 2017-2018 missing data. We have reworked the discussion of our tests on Lines 190-213 to more clearly explain the data and processes involved and more specific issues are addressed below.

From text and Table 1 it appears that the "root mean square difference" (which is the same as the standard deviation!) for daily average data is about 12 DU. So the 2 DU are probably for the monthly average data, but the 12 DU for the daily data should be mentioned here as well. (Assuming a Gaussian distribution, 67% of the data should be within +-1 standard deviation of the mean (which should be zero here by construction), 95% of the data within +-2 standard deviations, ...

Note: Table 1 (now Table 2) has been changed to display average differences with the Dobson as opposed to root mean square differences. This was done to maintain a consistent comparison metric across the paper.

We apologize again for the confusion that the 2 DU refers to a standard deviation. Table 1 (now Table 2) contains data about the raw satellite daily averages and is used as an introductory comparison of the individual instruments with the Dobson. The table also serves to explain our choice of using the satellite average. The caption has been updated to read *"(raw) daily measurements by GOME-2A, GOME-2B, OMI…"*

While I call this lack of clarity out here for line 21, it exists throughout the text, and needs to be fixed everywhere.

The phrase has been corrected throughout the text, and the discussion of the text has also been reworked.

line 39: Here it says "throughout the year", line 37 said that no data are available for May to July. What is true now?

It is true that data is not available for May to July. For clarity, we have changed the phrase to be: *"throughout the **measurement season**."*

line 45: delete "the" before "satellite"?

*The word "system" was accidentally left out of the sentence, it now reads "In the first decades of the satellite observing **system**…"*

line 50: replace "well tested" by "in place"?

*For clarity and concision, the phrase "With the advanced multi-satellite observing system now well-tested" was removed and replaced with "**Therefore**, we undertook the development…"*

Fig. 1: are those all measurements or daily averages? Please mention. Are the satellite data the original data from all satellites, or the adjusted data matching the Dobson?

*The values are daily averages and the satellite data plotted has not yet been adjusted. The caption has been updated to reflect this and now reads: "**Daily averages for** total column ozone measurements by Dobson instruments at Halley station (in black) overlaid on top of **all** available **(raw)** satellite daily averages (in red) from 2014-2019."*

Lines 79 to 104: Would be good to also give the size of the satellite ground pixels near Halley Bay for all the satellite instruments. In addition, I think it absolutely necessary to state which data version was used for each satellite, and where / from which URL the satellite data came from. For GOME2, for example, there are data from Uni-Bremen, from DLR / EUMETSAT, from RAL / ESA_CCI, ... A table of URLs and versions would help here.

*This is a good suggestion, and now provide the ground pixel size for each instrument in our revised discussion of each satellite instrument. In addition, we have created a table of URLs, versions, and sources for the satellite data (Table 1 in the revised paper).*

Line 85: "cross-calibrated" My understanding is that the current SBUV 8.6 version is not cross-calibrated between satellites, but relies on improved calibration at the radiance level for each satellite. Please check. Natalya Kramarova will know.

*The calibration of SBUV instruments for v8.6 is described in DeLand et al. (2012), which is now referenced in the revised paper. The calibration process included updates in calibrations and characterizations for each individual SBUV instrument as well as cross-calibrations of overlapping SBUV instruments. The cross-calibration process was an important step for producing consistent SBUV v8.6 ozone record, for more details on cross-calibrations see Sect. 3 in DeLand et al. (2012).*

Line 103: should be "polarization effects"

*The section on GOME, GOME-2, and SCIAMACHY was reworked to standardize the discussion of the satellite instruments. The line in question was removed during the process.*

Line 107: How were overpasses defined? Satellite foot-point within what distance? Same for all satellites?

*The criteria were not the same for all instruments.*

GOME-2, GOME, and SCIAMACHY: a weighted average from all footprints (pixels) available per orbit and within 100 km (300 km GOME/SCIAMACHY) of the station were defined as satellite overpass value. All instruments are flying in a near-polar orbit so that at many days several orbits fly over Antarctica. The daily mean overpass ozone was calculated from averaging over the overpassing orbits. The weights are given by sqrt(1-distance$^2$ [km$^2$]/max_distance$^2$). A line briefly describing this has been added to the revised paper: "**_Daily mean overpasses were calculated by averaging ozone columns from all ground pixels within 100 km (GOME-2) and 300 km (SCIAMACHY, GOME) of the station._**"

SBUV: the method for creating overpasses for SBUV is described by Labow et al. (2013, see Sect. 5 there). The overpass algorithm for the SBUV data has been created to return daily overpass values each day, even if the SBUV measurements are not directly overhead of the ground station. A box of 2° in latitude and 20° in longitude (large enough to encompass two orbits), is chosen around the ground station's location. The SBUV ozone measurements first interpolated along the orbital track with 0.5° step. Then the SBUV value at the station is calculated as a weighted 1/distance average using all points in the box. A line referencing Labow et al. (2013) has been added to the SBUV discussion.

OMI/TOMS/OMPS-NM: For the overpasses each day the single pixel most nearly co-located with the ground station is selected. At high latitudes a given ground location can be viewed from multiple orbits. In that case a pixel with very high optical path will be rejected in favor of one with a slightly poorer spatial coincidence but with a lower optical path.

OMPS-NP: The overpasses are based on the pixel closest to the station.

The following lines have been added to the discussion of the NASA GSFC instruments: "**_Overpasses for the TOMS, OMI, and OMPS-NP instruments are defined by selecting the single pixel most nearly co-located with Halley Station. In the case of there being multiple pixels available, a pixel with high optical path will be rejected in favor of one with slightly poorer spatial coincidence but lower optical path. For the OMPS-NP instrument, the pixel closest to the station is chosen._**"

line 127: Would it not be better to have Figure 3 and lines 155 to 160 right here in section 2.3? After all, the Figure shows the Δ-s that are discussed in lines 120 to 127?

We decided to keep Figure 3 under Results because it is the result of the methodology described in section 2.3. The discussion on the figure has also been expanded (L174-179, revised paper).

Line 128, Section 2.4: Would it not be clearer, to have section 2.2 here, after section 2.3. That way, you would have a more logical flow. a.) discuss Δ-s for individual satellites b.) discuss how you use all satellites to fill in, and how that looks for the different months.

We decided to keep the order of sections 2.2 and 2.3 because the calculation of the Δs uses the averaged data described in section 2.2.

Table 1: What is shown here? Differences between monthly averages, or differences between daily averages? From the numbers, around 12 DU, it looks like it was daily averageds. Were the satellite data Δ-adjusted or not? In April, that would make a large difference according to Fig. 3.

Note: Table 1 (now Table 2) has been changed to display average differences with the Dobson as opposed to root mean square differences. This was done to maintain a consistent comparison metric across the paper.

The differences are between the daily averages, which were then averaged across each month and in total. The satellite data has not been adjusted here. Section 2.2 (Data Analysis) now better explains this: "With all measurements and differences in the form of averaged daily values, data were categorized **and then averaged** according to their corresponding month and day of year (DOY)."

The caption for Table 1 (now Table 2) has also been changed for clarity, it now reads: "***Average absolute*** *differences in DU between the total column of O3 retrieved from the Halley Dobson instrument and those retrieved from the **(raw) daily** measurements by GOME-2A, GOME-2B, OMI, OMPS-NM, OMPS-NP, SBUV **averaged by month and in total** for the period from 2013-2018.*"

Line 143: Would the Δ-adjustment not take care of the Bass-Paur difference as well? Is it necessary to mention systematic biases here, since the filling-in method takes care of them anyways?

Although the Δ-adjustment would most likely take care of the Bass-Paur difference as well, we felt that not mentioning and correcting the systematic bias would misrepresent the performance of the GOME-2 instruments in Table 1 (original paper, now Table 2). Additionally, we believe that it makes sense to correct a systematic bias that is already known and explained.

Figure 4: Having Figure 4 so close to Figure 3 confused me (Are they now using monthly Δ-s again? Or daily? Or what?). I guess the only point of Figure 4 is to show that 2019 was very different from the other years. This does not become very clear here. The stars for 2019 are easy to miss in the Figure, and they do not have error bars. It would be helpful to have a clearer Figure, that points out 2019 in a legend in the Figure, not just in the caption.

(Addressed along with the next comment)

Also Figure 4: What are the error bars? Standard deviation of daily data or monthly data? Standard error of the mean? One or two standad deviations?

We apologize for the confusion. The purpose of Figure 4 is to show that 2019 was an anomalous year using monthly Δs. We have made the stars more visible, added a legend, and clarified what the error bars represent. The rewritten caption reads: *"Average Δ over all years (Fig. 2) excluding 2019 for each month with error bars (black).* **The monthly Δ values with the automated Dobson in 2019 (red) are much larger than other years. The error bars**

*represent the standard error of each satellite mean, combined in quadrature for each monthly bin."*

Lines 169 to 187, and Figure 5: I am confused. Does Fig. 5 show data, where 2003 to 2012 was the training period? Or which training periods were used to generate the data in the two panels of Fig. 5?

For tests done on 2013-2015, we pretended the Dobson data was "missing" for that period and each satellite was Δ-adjusted using Δs calculated from the rest of its available data, excluding 2019-2020. We have reworked the discussion of Figure 5 to clarify our test, particularly in Lines 190-202.

Lines 184, 185: I assume that the numbers are for the trained data? Please state the same numbers for the unadjusted satellite data. Only then you can conclude if the adjusted date are better, or not. Check consistency with numbers in abstract and conclusions!!

Thank you for this suggestion! When we calculated the average differences for the unadjusted satellite data, we found values of 6.5 DU for 1998-2002 and 4.6 DU for 2013-2015, which were much larger than the values for the adjusted data ($1.8 \pm 6.7$ DU and $1.1 \pm 6.2$ DU, respectively). The uncertainty represents the estimated training error, so there is no comparable value for the raw data. A few lines comparing the two have been added in the discussion of Figure 5 (L209-213).

We have also corrected the abstract and conclusion to say *"with **an average difference of 1.1 ± 6.2 DU**"* in order to be more consistent with the results.

Figure 6: The differences between the Dobson monthly means and the Δ-adjusted satellite data look rather large in 2019 and 2020, 5 to 10 DU. Is that consistent with the numbers given in lines 184, 185? Figure 4 shows that the 2019 Dobson data are flawed. Are the 2020 Dobson data flawed as well? Flaws of the Dobson data should be stated, and maybe even marked with different sysmbols in the Figure. How do the Δ-adjusted satellite data look in the other years? It would be good to plot the entire red time series.

Yes, we believe the 2020 Dobson data are flawed as well due to likely inconsistencies between the automated instrument and earlier data (Figure 4). The use of the automated instrument was continued in 2020. We have added the following sentence in the discussion of Figure 4: ***"Because the station continued to use the automated instrument in 2020, we treated the 2020 data as likely inconsistent as well and excluded it from our Δ adjustment."*** We also now note this in the caption of Figure 6 with the following added sentence: ***"Dobson data from 2019 and 2020 were filled in due to apparent inconsistencies between the automated instrument and earlier data."***

Because 2019 and 2020 were excluded from the Δ adjustment, the numbers given in lines 184 and 185 (original paper) are not inconsistent with Figure 6.

We decided not to plot the entire red time series along the Dobson because the primary focus of our study is to fill in gaps in the Dobson as faithfully as possible, not to investigate between the Dobson and satellite instruments. The purpose of Figure 6 is to present the now complete record.

Line 230: Are the 2 Dobson Units the average difference? Is that really relevant? In principle, the average difference should be zero, due to the $\Delta$-adjustment. Of course zero is not realized in every subset / realization of the data. Is not the standard deviation between Dobson and $\Delta$-adjusted satellite data a much more meaningful quantity, to show how well the two data sets agree?

(Addressed along with the next comment)

Also line 230: Check consistency with the numbers in abstract and in lines 184, 185. Please give (also) the standard deviations of Dobson minus $\Delta$-adjusted satellite data on the basis of monthly and daily means.

Yes, the 2 Dobson Units are meant to represent the average difference, but the line has been corrected to read: "average difference of $1.1 \pm 6.2$ DU for monthly averages," with the uncertainty now included. The average difference was not zero because this was the result of calculating $\Delta$s using the rest of the data and using it to fill in "missing" 2013-2015 data (explained above).

We felt that the statement *"we could fill in missing months with a high degree of fidelity"* should be backed up quantitatively, hence why the test results are included. The numbers should now all be consistent across the paper.

---

## Author Comment (AC2)

The authors would like to thank the reviewer for taking the time to provide feedback on "On the Use of Satellite Observations to Fill Gaps in the Halley Station Total Ozone Record." The insightful comments we received helped us understand how to improve on and better communicate the ideas presented in the paper. Nearly all suggestions were incorporated, and a line-by-line response can be found below (with author comments in blue):

**RC2: 'Comment on acp-2021-122', Anonymous Referee #2, 21 Apr 2021**

The authors use observations from multiple satellite instruments to create an ozone column dataset above the historically and scientifically significant Halley station in Antarctica, using the Dobson instrument as a calibration anchor. The result is a dataset that can fill in gaps due to a recent ice crack, check for calibration issues affecting the new Dobson data, and fill in gaps caused by other future geophysical or social disruptions.

**General comments**:

Overall, I enjoyed reading this work. The manuscript is well written and organized. Gaps in scientifically significant long-term datasets – such as the Halley Station's Dobson – are an important problem. The paper contributes usefully to this topic. I have a few questions and requests for clarification.

Thank you for the positive feedback!

The satellite instruments used in the study have a variety of measurement techniques, advantages and limitations. While it might not be necessary to do a deep dive into this, I think it is at least necessary to comment on the different spatial resolutions (and perhaps vertical sensitivities) and the spatial coincidence criteria used to define co-location with Halley Station.

We agree that this would be useful information to provide, and the revised paper discusses spatial resolution and spatial coincidence criteria in a more standardized format for all satellite instruments (L83-123).

Figure 1 is a key illustration of the datasets involved in this study – i.e., the adjusted average satellite and Halley Station Dobson. The figure is limited to 2013 – 2019, I assume because this allows features on the ~monthly timescale to be seen and because 2013 – 2015 was one of the time windows used to test the technique's ability to reproduce the Dobson. And 2017 – 2018 was the motivating gap in the Dobson's timeseries. Nonetheless, I would like to see the full timeseries of the two datasets. Figure 6 does some of this, but I believe it only inserts the satellite dataset into the Dobson gaps (?). That's useful, but I'd also like to see both fully plotted to get a sense of how closely they agree.

We apologize for the confusion. Figure 1 shows the Halley Station Dobson plotted alongside the **raw** satellite daily averages from all instruments. Its purpose in the introduction is to display the 2017-2018 Dobson gaps that are the focus of our paper as well as illustrate how the satellite instruments provide complimentary information during the gap periods. The caption has been reworded to say: "*Figure 1: **Daily averages for** total column ozone measurements by Dobson*

*instruments at Halley station (in black) overlaid on top of **all** available **(raw)** satellite measurements (in red) from 2014-2019.”*

As requested, we have added additional timeseries plots for the complete time series of the available satellite data versus Halley in the supplement (new Fig. S1-S3).

I am not entirely comfortable with the conclusion that the adjusted satellite average reproduces the Halley Station Dobson to within about 2 DU. I don't think it was sufficiently stated that the satellite reproduction of the Halley Dobson data varies seasonally. In Figure 5, some months show very close agreement; others show much larger differences, e.g., ±15 DU. 2 DU is the apparent result of averaging large positive differences and negative differences across the year. In addition, while there are similar annual patterns, there are also notable differences between the years shown.

Yes, it was incorrect of us to conclude that the adjusted satellite average reproduces the Halley Station Dobson *“to within an average of 2 Dobson units”* for the reasons you mentioned. We have re-evaluated the numbers carefully, and rewritten the sentence to represent our results more accurately, and it now reads: *“**Comparisons to the Dobson** suggest that our method reproduces monthly ground-based total ozone values **with an average difference of 1.1 ± 6.2 DU for the satellites used to fill in the 2017-2018 gap.***”*

In addition, the time periods chosen for the test have a particular combination of satellite instruments that are being used to compare with the Halley Station Dobson. What were the results of the comparison between the adjusted satellite average and the Dobson for wider time periods? How well do these test cases, 1998 – 2002 and 2013 – 2015, generally represent the physical conditions and satellite datasets available to fill other gaps and future gaps?

This is a good point. Given that our study primarily focuses on developing a sound methodology for filling in the 2017-2018 gaps, we chose the 2013-2015 testing period because the instruments in operation are the same as the ones in 2017-2018 and would presumably be used in future efforts (GOME-2A, GOME-2B, SBUV, OMI, OMPS-NM, and OMPS-NP) until they cease operations. Unfortunately, we are unable to test on a wider time period for this particular set of instruments because the majority of them are only available from around 2012-2020. If we widened the training period, we would run the risk of not having enough years to "train" the Δs on. A similar issue arises with the four instruments available in 1998-2002.

We understand that the decisions made in our tests as well as the methodology may be confusing, so we have rewritten the discussion to clarify it as follows:

*“Consequently, we chose to test the method for the years 2013 to 2015 by pretending data for those years did not exist and characterizing the monthly Δ values averaged over those years **using the rest of the available data for the GOME-2A, GOME-2B, OMI, OMPS-NP, OMPS-NM, and SBUV instruments.** To examine the performance of our method during periods when there were fewer available instruments, we also tested on 1998-2002 using **data from GOME SCIAMACHY, SBUV and EP/TOMS instruments. The range of available***

*data for each instrument can be found in Figure 2. The training period for each instrument is the available range after excluding the years being tested (and 2019-2020)."*

The approach described in the paper, and the resulting dataset, is useful, but I think care needs to be taken in asserting the accuracy in reproducing the Dobson shown here – especially since particular months of the year are often of more significant interest to the study of ozone chemistry than annual averages.

Thank you again for the positive feedback. We agree that particular months of the year are often of more significant interest to the study of ozone chemistry, and our results (i.e., Table 1, now Table 2) suggest that the satellite average does not exhibit anomalously high differences during the months that comprise the ozone hole season. However, we have corrected our conclusions to be more representative of our results—as mentioned above.

Early in the paper (section 2.2) it is stated that both absolute and relative differences are computed. The paper then focuses on the absolute differences. Given the annual pattern to the absolute differences, I'm curious to know what the results of the relative differences were? Do percent differences show the same seasonality as the absolute differences? Would constructing a delta adjustment on the basis of a relative difference or SZA remove some of the annual pattern in the difference?

Yes, relative differences were computed but showed the same seasonality as the absolute differences. Early on in our research process, we examined solar zenith angle dependence as well and determined that the delta is not simply dependent on that. We decided that this topic should be a focus of a later paper, and our satellite and BAS coauthors are aware of the issue. However, we agree that the issue is important to mention in the paper, and a line has been added in section 2.3 (Delta Characterization and Adjustment). It now reads as follows:

*"…the Δ value for each day of year is the average of the absolute differences between each satellite and Dobson for that day of year, across all years in each satellite series. **Relative differences were also computed but displayed the same seasonality as absolute differences**."*

Figures:

It would be helpful to expand the width of the figures to fill the width of the page. Figure 1, for example, would benefit from this since it can be difficult to see the structure in the data and the comparison between the Dobson and satellite measurements. This is important for understanding the work being described. Figure 3 as well.

We agree with the suggestion. All figures have been widened to better fill the width of the page.

Figure 4: why is there no monthly value for August (month 8) 2019? Caption title could add that 2019 is with the automated Dobson so that this context stands alone in the figure without the text. I'm assuming the error bars are the standard error average? Is the August error bar larger because there are fewer days being used (polar night limitation)?

Thank you for catching this error. The value for August 2019 was cut out of the original figure and the axes limits have now been corrected to include the data point. To make the goal of the figure more clear, we have also changed the caption to read: *"Average Δ over all years (Fig. 2) excluding 2019 for each month with error bars (black).* ***The monthly Δ values with the automated Dobson in 2019 (red) are much larger than other years. The error bars represent the standard error of each satellite mean, combined in quadrature for each monthly bin.*"

It is likely that the error bars in August and April and larger due in part to the polar night limitation; we have added a line mentioning this limitation in section 2.2 (data analysis). The line is: "***Months directly bordering the polar night (April and August) contained fewer data points when computing monthly averages.***"

Figure 6: has tick labels that are too small to easily read; they are notably smaller than other figures.

We have fixed the axes font size.

Tables:

Table 1 is awkwardly split across page 5, which has the caption and titles, and page 6, which has all the values. Please note this to the typesetter and check that the proofs correct this.

Thank you for pointing this issue out, the table should no longer be split in the revised paper but we will make a note to the typesetter.

Could be informative to add a column for the delta-corrected satellite average and/or the delta adjustment.

We decided to not introduce the Δ-adjusted satellite average in Table 1 (now Table 2) in order to keep the purpose of Table 1 (now Table 2) as it was: comparing the various (raw) satellite data and justifying our use of the satellite average to fill in the gaps.

Abstract:

In a few cases, it might be helpful to the reader to add specifics. For example,

"by… adjusting overpass data" – adjusting how?

The line originally read as *"bias-correcting overpass data,"* but we were advised by the editor to remove mentions of "bias" throughout the text—hence why we refer to "Δs" and "Δ-adjustment." The specifics of our adjustment are explained in the Methods section of the text.

"Tests suggest that our method…" – what tests? Or rephrase to say "comparisons to ___ suggest that our method…"

We have changed the sentence to be more descriptive, it now reads: *"**Comparisons to** the Dobson suggest that our method reproduces monthly ground-based total ozone **values with an average difference of 1.1 ± 6.2 DU for the satellites used to fill in the 2017-2018 gap.**"*

"… our approach improves on the overall performance…" – what does overall performance refer to? Accuracy of the measurement? Comparison results? Completeness? Also want to be careful because the goal of the paper, as stated at P3L57-58, is not a high-performance dataset / "most accurate" dataset, but "to reproduce what the Dobson instrument would have observed".

This is a good point. The phrase *"our approach improves on the overall performance"* has been changed to: *"our approach **more closely reproduces the Dobson measurements** than…"*

"… there was a significant difference between the two." – I'd suggest being more quantitative. What was the difference? This also brings up a question of what the authors consider to be a threshold for good agreement?

Since the primary focus of our paper is to fill in the Halley Dobson record and reproduce total ozone values as would have been measured by the Dobson, we decided not to delve too deep into the topic of agreement between the different satellite instruments and the Dobson. Throughout the text (i.e., L251-252) we suggest that it could be a topic of future study using our method and datasets).

We agree that the term *"significant difference"* is too vague because of its quantitative implications, so we have replaced it with: *"there **were apparent inconsistencies** between the two."* This statement is backed up in the text by Figure 4.

**Specific comments**:

P2L1: A good paragraph starting sentence. But, suggest rephrasing "now", since the interruption being discussed was a few years ago.

This is true. The sentence has been rephrased to say: *"**In 2017**, this remarkable record **was** interrupted."*

P2L49: (warning: pedantic point) Most readers will understand what is meant by "With the advanced multi-satellite observing system now…" but the wording here might suggest that the current (and past) suite of satellites are part of a coherent and coordinated "system". We have a great scientific community and missions are chosen to provide complementary coverage. But I'm not entirely convinced the satellites used in this study, from different agencies, eras, and designs, are "an observing system".

That is a good point. For clarity and concision, the phrase *"With the advanced multi-satellite observing system now well-tested"* was removed and replaced with *"**Therefore**, we undertook the development…"*

P4L79: missing parenthesis before the Munroe citation.

Thank you—fixed.

P4L76: "…some also have spectral information at other wavelengths." – what instruments and what other wavelengths?

The spectral information at other wavelengths was not relevant to our study and the segment has since been deleted.

P4L80: Suggest re-wording "spin off of the" – it isn't clear what is meant by "spin off".

The entire section on the GOME and SCIAMACHY satellites have been revised. The word "spin off" is not used anymore.

P4L81: "… of a somewhat improved…" – "somewhat" is hard to interpret. What was improved (or not) that is notable here?

(addressed along with the next comment)

P4L79:82: why put this information about GOME here when the details of this instrument are four paragraphs below? Doesn't serve the flow.

Thank you for pointing this out. We have reorganized the paragraphs on the satellite instruments to be in a more standardized format. The information about GOME is now consolidated in Lines 83-91.

Because the line in question was not relevant to our study (P4L81, original paper), it was removed when the section was reorganized.

P4L83: SBUV acronym used before defined. Need to carefully define, given it is an instrument and set of instruments.

The SBUV acronym is now defined, the revised sentence reads: *"The Solar Backscatter Ultraviolet (SBUV) record…"*

**P5L111**: It is important to be clear what has been done to define the spatial coincidence criteria for the satellites and Halley Station.

Spatial coincidence criteria is now discussed in Lines 88-91 (GOME, GOME-2, and SCIAMACHY), Lines 98-99 (SBUV), and Lines 115-118 (TOMS/OMI/OMPS-NM/OMPS-NP).

P5L117: "… due to unusually high differences…" – what were the differences.

Once again, we decided not to delve into the topic of what a threshold for agreement between the Dobson and satellite instruments would be. We have rewritten the sentence as follows: *"lunar Dobson measurements from August 24th, 2015 were excluded due to **obviously anomalous** differences."*

P5: Section 2.3: You state that the delta value for each DOY is averaged across all years in each satellite series. Was there any trend or interannual variability in the differences before combining all years?

That is a good point. The uncertainty in the $\Delta$-characterization and adjustment due to interannual variability in the differences is captured by our estimated training error. We now address this in Methods—here is the revised line: *"Uncertainty for the delta-adjustment of the satellite average was calculated by combining, in quadrature, the standard error of the mean for each satellite **and accounts for interannual variability."***

We have also added a phrase explaining the uncertainty metric more clearly in the Results section: *"1.8 ± **an estimated training error of** 6.7 Dobson units (DU)" (L208).*

P6L149: Why is April different?

(addressed along with the next comment)

P8: The observed differences are typically larger than the average delta, which is reduced by large positive and negative values averaging out. If this pattern is due to SZA-dependence, would a bias correction based on (or including) SZA rather than DOY produce a useful result?

For an unknown reason, $\Delta$ does not seem to follow a simple solar zenith angle dependence, as evidenced by the much higher values computed in April than in August at similar solar zenith angles. We choose not to speculate on this in our paper, but it is the reason that we chose to do a bias correction based on DOY rather than SZA. We have reworked the discussion of Figure 3 to explain this component of our reasoning and added the following sentences:

*"**Additionally, $\Delta$ does not follow a simple solar zenith angle dependence. Values differ between the onset and end of the polar night for days with the same solar zenith angle, as evidenced by the larger $\Delta$s in April versus August. Therefore, we chose to characterize $\Delta$ by day of year rather than zenith angle."***

P8: If there was an unusually high or low level of ozone abundance, will the absolute delta correction sufficiently reproduce the Dobson? Would a relative (%) adjustment scale better?

The agreement at Halley appears to scale with ozone abundance. We are not sure why this is, but the issue does not seem to persist when comparing the satellite instruments to the Syowa station. We have alerted our BAS coauthors to this issue but decided this should be the focus of a different paper, as our primary goal is to reproduce what the Dobson would have seen—not discuss the accuracy of the Dobson.

P9L190: suggest adding a comment about calculating averages for the months that border polar night, where there will be a reduced number of days contributing. I don't know offhand when Halley Station's latitude enters/exits polar night. That might be worth mentioning somewhere since it is relevant to data collection and chemistry.

We agree this is worth mentioning. We have added the following sentence to our discussion of the averaging in section 2.2: *"**Months directly bordering the polar night (April and August) contained fewer data points when computing monthly averages.**"*

*More information about the measurement season at Halley Station has also been added to section 2.1 (Data). It now includes the lines:* *"**Halley solar data typically end on April 16th as the sun retreats for polar night, and resume on August 27th. There are also some limited lunar measurements.**"*

P12L223: "Larger difference" – how large? Since this is an important finding, it would be good to state this. Hopefully this prompts investigation into why there might be differences between the measurements.

Figure 4 was developed so the reader could judge this for themselves for their season of interest. We revised the paper to: "*we found that the preliminary computed data from the automated instrument in 2019* **had apparent inconsistencies with the earlier data taken with the manual Dobson** *when compared to the satellite average* **(see Fig. 4)**."

P13L227: were any of the satellites used in this study validated using the data collected at Halley by the Dobson?

The WFDOAS (GOME, SCIAMACHY, GOME-2) approach has been validated using Halley Bay data (Weber et al. 2005, Orfanoz-Cheuquelaf et al., 2021) but the rest of the instruments have not. This is now mentioned in the text:

*"**The WFDOAS approach was validated using Halley station data as reported in Weber et al. (2005) and Orfanoz-Cheuquelaf et al. (2021).**"*

*"**None of these instruments [TOMS, OMI, OMPS-NM, OMPS-NP], as well as SBUV, were validated with Halley station data.**"*

P13: There was quite a bit of discussion about the DOY data but the conclusions focused on the monthly averages. Why not include a plot of the DOY comparison results as well and comment on their comparability to the monthly results?

The results focus on the monthly averages because they are the easiest to visualize and tabulate in the paper. We felt that the DOY plots would be too messy to draw any useful conclusions. With respect to the tests, the limitations of the testing periods (mentioned above) would only allow us to make comparisons for a maximum of three datapoints for each DOY.

We agree that the filled DOY record should be provided, and we have made the data available at: https://www.ssolomongroup.mit.edu/toolsandproducts

---

## Author Response (AR2)

Dear editor,

We appreciate your careful consideration of our manuscript and agree with the additional suggestions you have made. Please find our response to each comment in blue.

Comments to the Author:

Dear authors,

I am pleased to inform you that your paper is published in ACP after consideration of the following technical corrections:

- The acronyms of the satellite instruments should already be introduced in line 75 on page 2 (first occasion where these are mentioned).

The introduction of each satellite instrument now reads:

*"the following eleven instruments (Fig. 2): GOME (**Global Ozone Monitoring Experiment**), GOME-2A, GOME-2B, SCIAMACHY (**SCanning Imaging Absorption spectroMeter for Atmospheric CartograpHY**), SBUV (**Solar Backscatter Ultraviolet**), N7/TOMS (**Total Ozone Mapping Spectrometer on Nimbus-7**), M3/TOMS (**Meteor-3**), EP/TOMS (**Earth Probe**), OMI (**Ozone Monitoring Instrument**), OMPS-NM (**Ozone Mapping and Profiler Suite, Nadir Mapper**), and OMPS-NP (**Nadir Profiler**)."*

- Fig 4 caption: Why "larger"? This is not clear if I look at the figure and see lower values. Please be more clear here.

We apologize for the confusion. The caption has been rewritten to say *"The monthly Δ values with the automated Dobson in 2019 (red) **have larger magnitudes than Δs in** other years."*

- P10, L208: I would suggest to write this text part as follows: " .....with and average and an estimated training error of 1.8+/-6.2 Dobson Units (DU)...."

The change has been made in the text.

- P12, L230 and 233: Here a p-value is given. What is the meaning of this value. This should be clarified in the text.

We now discuss the meaning of the p-value and have added the following line: *"**A low p-value (p ≤ 0.05) for the regression indicates that the trend is unlikely to have occurred by chance**."*

- Figure 6 caption: "were refilled"? I would rather say you mean "were replaced"?

The caption has been changed to say *"**replaced**"* instead of *"filled."*

- In the manuscript you mention several times that the 2019/2020 data is inconsistent with the previous data record due to the switch from manual to automated operation. Why these data becomes different just due to the switch manual to automated is not clear and it would be good if you could give a reason. In the conclusion you mention "calibration issues". Is that the cause?

Yes, we now provide an explanation in the text, with the discussion of Figure 4: *"This indicates likely inconsistencies between the automated instrument and earlier data.* **Every Dobson instrument must be carefully calibrated to ensure accurate data; the calibration process for the automated instrument has not yet been completed.**"